# Exploring the Inflammatory Pathogenesis of Colorectal Cancer

**DOI:** 10.3390/diseases9040079

**Published:** 2021-10-30

**Authors:** Ahamed A Khalyfa, Shil Punatar, Rida Aslam, Alex Yarbrough

**Affiliations:** Department of Gastroenterology, Franciscan Health, Olympia Fields, IL 60461, USA; Shil.punatar@franciscanalliance.org (S.P.); Rida.Aslam@franciscanalliance.org (R.A.)

**Keywords:** colorectal cancer, inflammation, exosomes, obesity and diet, microbiome, mycobiome

## Abstract

Colorectal cancer is one of the most commonly diagnosed cancers worldwide. Traditionally, mechanisms of colorectal cancer formation have focused on genetic alterations including chromosomal damage and microsatellite instability. In recent years, there has been a growing body of evidence supporting the role of inflammation in colorectal cancer formation. Multiple cytokines, immune cells such T cells and macrophages, and other immune mediators have been identified in pathways leading to the initiation, growth, and metastasis of colorectal cancer. Outside the previously explored mechanisms and pathways leading to colorectal cancer, initiatives have been shifted to further study the role of inflammation in pathogenesis. Inflammatory pathways have also been linked to some traditional risk factors of colorectal cancer such as obesity, smoking and diabetes, as well as more novel associations such as the gut microbiome, the gut mycobiome and exosomes. In this review, we will explore the roles of obesity and diet, smoking, diabetes, the microbiome, the mycobiome and exosomes in colorectal cancer, with a specific focus on the underlying inflammatory and metabolic pathways involved. We will also investigate how the study of colon cancer from an inflammatory background not only creates a more holistic and inclusive understanding of this disease, but also creates unique opportunities for prevention, early diagnosis and therapy.

## 1. Introduction

Colorectal cancer (CRC) is the third most commonly diagnosed cancer in the United States. According to the National Cancer Institute, the estimated number of new cases of colon cancer for 2021 is 149,500 with the number of deaths estimated at 52,980 [1]. Common risk factors for colorectal cancer include hereditary history, age, diabetes, obesity, smoking history, and alcohol dependence. Although 25% of cases occur in patients with a family history of CRC and 10% occur in hereditary colorectal syndromes, almost 65% of cases occur sporadically in average-risk individuals [2]. Recent advancements in molecular techniques have improved the understanding of the pathogenesis of sporadic and hereditary colorectal cancer syndromes. These advancements have allowed us to classify the development of colorectal cancer into two separate, genetically independent pathways: the conventional pathway, including the adenoma-to-carcinoma sequence, and the serrated pathway [3]. The conventional pathway is characterized by genetic mutations in APC, KRAS and p53, while the serrated pathway is known for alterations in BRAF, CDKN2A and MLH1 methylation [3].

Although the adenoma–carcinoma sequence and serrated pathways are widely accepted by the scientific community, there has been increasing evidence regarding the role of inflammation in colorectal cancer. Inflammatory bowel disease (IBD), including ulcerative colitis and Crohn’s disease, has long been associated with an increased risk of colorectal cancer. Chronic inflammation caused by *Helicobacter Pylori* has also been associated with a moderate increase in colorectal cancer risk [4]. Primary sclerosing cholangitis, an inflammatory hepatobiliary disease, also increases the risk of colorectal cancer by approximately fourfold compared to patients with ulcerative colitis alone [5]. Interestingly, the APC, KRAS and p53 mutations have been associated with colitis-related CRC, albeit less frequently when compared to sporadic cases of CRC [6,7]. The hMSH2 mutation, which has been implicated in hereditary nonpolyposis colorectal cancer, was reported to be more frequent in ulcerative colitis patients who developed CRC than in patients who did not develop CRC [8].

Even though there are similarities between sporadic CRC, hereditary colorectal syndromes and colitis-related CRC on the genetic front, differences have been established regarding their bio-pathological features [9]. Specifically, compared to patients with sporadic CRC, patients with ulcerative colitis related-CRC were found to have a higher proportion of multiple lesions, a higher number of superficial and invasive lesions, and a higher proportion of signet ring cell or mucinous carcinomas [9]. Furthermore, in comparison to the dysplastic precursor of adenoma associated with sporadic CRC, the dysplastic precursors for colitis-related CRC have been found to be polypoid or flat, localized, diffuse, or multifocal, further suggesting an “inflammation-dysplasia-carcinoma” sequence in colitis-related CRC as opposed to the “adenoma-carcinoma” sequence of sporadic CRC [10]. Given the morphologic differences between these two pathways, much research has been dedicated to delineating the inflammatory pathways involved in the development of colitis-related CRC. More specifically, inflammatory signaling pathways such as nuclear factor kappa B (NF-*κ*B), IL-6/STAT3, cyclooxygenase-2 (COX-2)/PGE_2_, and IL-23/Th17 have not only been identified in the propagation of colitis-related CRC, but also in non-colitis related CRC [11,12,13,14,15]. Furthermore, research into inflammation and colorectal cancer development has given us great insight into the tumor microenvironment (TME), one that consists of stromal cells, the extracellular matrix, blood vessels, and immune cells, including tumor-associated macrophages [16]. Importantly, the study of the impact of inflammatory cells, such as tumor-associated macrophages, on CRC development has augmented our understanding regarding other proinflammatory conditions such as obesity and the microbiota and their respective roles in CRC development.

Here, we will explore the impact of obesity and diet, smoking, diabetes, the microbiota, and intestinal fungi on the development of CRC, focusing on the inflammatory and metabolic pathways that lead to a pro-neoplastic milieu. We will also discuss exosomes, including what they are, and how an understanding of these particles not only advances our knowledge of inflammation and colon cancer, but also presents us with an opportunity to use them as diagnostic and therapeutic markers. We will also explore how an understanding of the inflammatory mechanisms associated with colorectal cancer can lead to the early detection and prevention of this deadly disease.

## 2. Obesity and Diet, Smoking, Diabetes in Colorectal Carcinoma

According to established accounts, colorectal carcinoma is considered the second most predominant obesity-related cancer [17]. Studies have demonstrated an increase in the prevalence of obese individuals, accounting for over 15% of adult women, and 30% of adult men, with trends demonstrating that obesity in adolescence is associated with the development of colon cancer later in life [18]. With this, we will discuss the role of obesity in colorectal carcinoma, as manifested in inflammatory, metabolic and signal transduction pathways.

In recent years, we have developed a refined understanding regarding the intrinsic endocrinological properties of adipose tissue in the regulation of adipokines and biologically active molecules, which lead to pleiotropic effects such as angiogenesis and inflammation [19]. Leptin is one such hormone, found in greater prevalence in CRC [20]. Current mechanisms suggest that Leptin plays a role in colorectal carcinoma cells through PI3K/Akt and mTOR pathway regulation and invasion, as well as those implicated in the JAK2, STAT3, and NF-kB pathways [21]. Leptin also leads to the upregulation of proinflammatory genes such as IL6, IL1-beta, and drives VEGF-induced angiogenesis [21]. Epidemiological studies have shown an increased risk of colon cancer in people with high levels of leptin [22,23]. Other studies highlight an important interaction between leptin and soluble leptin receptor in the development of CRC [24]. Although further research is necessary to confirm the specific mechanisms by which leptin is implicated in CRC development in obese individuals, these studies suggest potential novel approaches for refined risk stratification using biological markers, and the early diagnosis and prevention of CRC. Another hormone which is decreased in higher levels of adipose, adiponectin, plays a role in the reduction in chronic inflammation via mechanisms preventing goblet cell apoptosis [25]. This is created as adiponectin reduces leucine-rich repeats containing G protein coupled receptors (Lgr5^+^), leading to less KRAS mutant colorectal carcinoma cases [21]. Adiponectin also demonstrates a role in downregulating the mTOR pathway, one that is exploited by cancer cells in order to produce unregulated cellular proliferation [26]. In vitro studies have demonstrated the role of adiponectin-suppressing angiogenesis in colon cancer with the regulation of STAT3, VEGF and cell proliferation [27]. Obesity is associated with a higher concentration of insulin and insulin-like growth factors 1 (IGF-1) from the pancreas but IGF-1 is also derived from the mesenchymal cells around intestinal stem cells [28]. Higher insulin and IGF-1 levels independently raise the risk of colorectal carcinoma, with a corresponding increase in the risk of proliferation of intestinal stem cells [28]. The hormone Ghrelin, considered in conjunction with Leptin, has been found to play a role in the growth hormone and IGF-1 axis, with dysregulation and proliferation leading to upregulation of the Akt/PI3K pathways in carcinogenesis [26].

Obesity also has direct inflammatory implications for CRC cancer formation, independent of hormonal mediation. For example, obesity has been demonstrated to lead to the proliferation of intestinal progenitor and stem cells through macrophage and dendritic cell activation [29]. With this, there is release of cytokines such as IL-22, IL-6, IL-17 and TNF-alpha [28]. Studies have demonstrated that obesity leads to an adaptive immune response elevating CD8-induced cytotoxic T-cell reactions, reducing regulatory T cells and the recruitment of M1 macrophages in inflammation [30] (see Figure 1).

Another pathway between obesity and CRC formation is notch signaling. Notch signaling is a conserved pathway involved in adipose related thermogenesis and energy generation [31]. It has been implicated in evolutionary roles in cell proliferation and differentiation [31]. Notch signaling has higher levels of activation in colorectal cancers with the initiation of cancers, but not the progression [28]. It has also been highly expressed in adenomas and leads to increased chemoprevention and decreased sensitivity to therapy [28].

With newer evidence demonstrating the role of obesity and gut microbiota in influencing potential colon-cancer-causing mechanisms, attention has also been drawn to another modifiable component in the cycle, the human diet. Several mechanisms and dietary constructs have been studied regarding their role in cancer formation. While there has been an increase in studies demonstrating higher cancer risk with specific food groups, here we will consider the mechanisms by which these increase inflammatory, anti-inflammatory and oncogenic features. The role of short-chain fatty acids, and the lesser studied roles of processed meats and polyphenols will be analyzed.

### 2.1. Short Chain Fatty Acids

From a macroscopic lens, general recommendations are often made to follow diets such as the Mediterranean diet while increasing dietary fiber intake. The Mediterranean diet encompasses lower amounts of meat with an enhanced intake of vegetables, fruits, and nuts [32]. When coupled with the role of fiber, these collectively play significant roles in the gut microbiota, specifically the protective effects of the metabolites that form, namely short-chain fatty acids [33].

Several studies have demonstrated the mechanisms by which fiber reduces colorectal carcinoma development. Dietary fiber fermentation reduces fecal pH, decreasing bacterial carcinogens from bile metabolism [34]. This leads to reduced colonic transit time and colonocyte exposure to carcinogens [34]. The major microbial metabolites from the bacterial fermentation of dietary fiber and whole grains are short-chain fatty acids, primarily acetic acid, propionic acid, and butyric acid [35]. These metabolites are absorbed through passive diffusion into colonocytes and enter the citric acid cycle to generate energy [35]. Short-chain fatty acids work in two main pathways, with the inhibition of histone deacetylases and activation of cell surface receptors [36]. Under conditions of intestinal inflammation, lower levels of short-chain-fatty-acid-producing bacteria are found in the intestinal mucosa, promoting dysbiosis and proinflammatory cytokines such IFN-gamma, IL-17, IL-1beta, and TNF-alpha [36]. Butyrate induces apoptosis in colorectal cancer cells via the histone deacetylase inhibition pathway and with activation of the Fas-receptor-mediated death pathway [37]. High butyrate levels modify microRNAS as well as the MYC oncogene, playing roles in decreased cell proliferation and angiogenesis [37]. Short-chain fatty acid receptors, namely FFAR2/FFAR3, have also demonstrated tumor-suppressive effects [35]. FFAR2 allows for mediation of the butyrate-induced histone deacetylase pathway and reductions in the cAMP-PKA-CREB and Wnt pathways, which are known to promote carcinogenesis [35].

### 2.2. Polyphenols/Red Meats

Increased processed and red meat consumption have been found to lead to an over 20% increase in colorectal cancer prevalence [38]. This is mainly from dietary components such as heme and arginine with potential for mutagenesis and inflammation [39]. Heme iron from meat products has been shown to cause cellular destruction via reactive oxygen species and the inhibition of colonocyte apoptosis, leading to carcinogenesis [38]. Meat products also lead to increased nitrosamines, which are particularly carcinogenic [37]. Heme iron enhances lipid peroxidation, increasing bacterial production of aldehydes and enhancing genotoxic effects [34]. Arginine, used as a precursor to polyamines, increases cell proliferation [38]. Collectively, these compounds provide insight into how red meat leads to the activation of proinflammatory and pro-carcinogenic pathways. Given that colorectal cancer risk is governed by both genetic and environmental factors, recent research has focused on delineating these gene-by-environment interactions as potential determinants of CRC development. Gene-by-environment is based on the premise that, as there is an increasing number disease-associated alleles with high and low penetrance, we can test whether environmental factors modify allele penetrance [40]. With respect to the gene-by-environment interactions studied for processed meats and CRC risk, some evidence suggests a modifying effect of the rs4143094 genetic variant on the *10p14/GATA3* gene [41]. Although the specific mechanisms are not completely understood, given that *GATA3* has been shown to be instrumental in T cell development, one potential explanation for this is that the decreased expression of *GATA3* and decreased immunogenicity can increase CRC risk in the setting of unopposed inflammation caused by processed meats [42]. On the other hand, polyphenols, such as resveratrol and quercetin, found in wine, tea, coffee, vegetables and fruits, have some protective effects in tumor and oncogene suppression [43]. These influence pathways involved in microRNA, and TGF-beta1 transcription, and induce pathways with E2F3 and Sirt1 genes, which promote apoptosis in colonic cells [43].

### 2.3. Other Associations between Obesity and CRC

While the aforementioned sections regard the role of short-chain fatty acids as well as the implications of diets with an excess of red meat, attention has also been directed towards all-cause obesity in the pathogenesis of colorectal carcinoma. As mentioned previously, obese individuals have a higher prevalence of colorectal cancer formation; therefore, attention has been directed towards the fat mass and obesity (FTO) pathway [44]. In this genotype, an FTO protein is linked to fat mass in humans and is predominantly located in the hypothalamus, with control over satiety. This protein affects the intake of food overall as opposed to energy expenditure [44]. Interestingly, this purports the notion that, regardless of dietary composition, genetic changes influencing excess food intake may play a role in obesity and, therefore, colorectal pathogenesis.

While genetic factors have been studied in carcinogenesis, the augmentation of these factors is also dependent on region. For example, it has been shown that Japanese immigrants moving to Hawaii will acquire the same risk of colorectal cancer, attributed to diets rich in the aforementioned food groups lacking in non-refined grains and low in fruits and vegetables [45]. High-fat diets have been shown to lead to a higher development of colorectal carcinoma and this is partly due to the lack of protective elements found in fruits and vegetables [44]. Phytochemicals are micronutrients that are synthesized from plants and play a role in the anti-inflammatory and antineoplastic effects that prohibit carcinogenesis [46]. While certain food groups have demonstrated higher rates of carcinogenesis, the lack of food groups such as fruits and vegetables involved in the genetic factors leading to obesity create the notion that, with and without diet, obesity plays a role in increased carcinogenesis.

### 2.4. Smoking and Diabetes

In consideration of the lifestyle modifications that also play an adjunctive role in carcinogenesis, studies have now developed a further understanding of the role of tobacco smoking in colon cancer development. While studies demonstrate that men, on average, smoke more often than women, the smoking-related colorectal cancer risk is comparable between sexes, with some studies going so far as to establish patterns of higher prevalence on right- versus left-sided cancers [47]. Current studies have divided pathogenesis into inflammatory responses and modifications in the gut microbiome due to smoking. Inflammation caused by smoking has demonstrated changes leading to altered epithelial mucin composition, which, in turn, disrupts mucus production and enhances the inflammatory response in the gut [48]. Moreover, there is evidence that cigarette smoking leads to M2 macrophage polarization and release of IL-1 IL-10, IL-6, IL-8, IL-17, IFN-γ TGF-β1 and TGF-β2, as well as activation of JAK2 and STAT3 [49,50,51]. Regarding the microbiome, smoking has demonstrated a negative correlation with the diversity of the gut microbiome [48]. A higher prevalence of *Clostridium* and *Bacteroides* have been identified in fecal samples of smokers, with lower numbers of *Lactococcus*, *Ruminococcus*, and *Enterobacteriacae* [48]. When linked with diet, smoking leads to lower levels of *Bifidobacterium*, which, in turn, reduces the production of protective short-chain fatty acids [45]. Studies have also demonstrated that smoking cessation at least partially revives the diversity of the gut microbiome, which may lead to further emphasis on the importance of smoking cessation in colon cancer prevention [52].

Attention has also been brought to the role of diabetes in colorectal carcinoma development, through the lens of inflammation and microbiome modification. Strong links have been established between colorectal cancer and diabetes, with studies also demonstrating worse outcomes for patients who have both diabetes and CRC [53]. Current studies have established colonic sensory and mechanical changes in diabetes mellitus such as delayed transit time, which alters the gut microbiome composition [54]. Diabetes mellitus induces histomorphological changes, changing gut microbiome function and composition [54]. This affects intestinal immunology, mucosa integrity and smooth muscle function [54]. Inflammation is a key component of diabetes mellitus, with inflammasome regulation of gut microbiota and the responses of epithelial cells leading to tissue injury, favoring the development of colonic cancer [54]. More specifically, NF-κB has been found to be a key regulator in the crosstalk between diabetes, inflammation and CRC through the upregulation of IL-6, IL-1 α and TNF α [55]. Furthermore, it promotes epithelial-to-mesenchymal transition through JAK/STAT pathway activation [55]. Interestingly, TGF-β has also been identified as a key component in the crosstalk between diabetes, inflammation, obesity, and CRC. Consistently elevated glucose levels in type 2 diabetes mellitus leads to the upregulation of TGF-β, which subsequently leads to M1 macrophage recruitment, B cell apoptosis, reactive oxygen species, and adipocyte hypertrophy [56]. TGF-β also leads to the activation of IL-17, producing Th-17 cells, which play an important role in the pathogenesis of colitis [57]. Given this intricate inflammatory crosstalk, future directions and surveillance have also been considered in attempts to combat both colorectal cancer and diabetes. Recent studies have considered the role of HMGCoA reductase inhibitors, renin-angiotensin aldosterone system manipulation, vitamin-D-receptor activators and anti-inflammatory agents in simultaneously targeting colorectal cancer and diabetes [53].

## 3. Microbiome

The field of microbiome and colon cancer has produced a wealth of knowledge in recent years. Multiple strains of bacteria have been directly linked to colon cancer pathogenesis including *Fusobacterium Nucleatum*, *Streptococcus gallolyticus*, *Bacteroides fragilis*, *E coli*, *Enterococcus faecalis* [58]. Several recent review articles discuss different aspects of the microbiome as they relate to colon cancer such as biofilm formation, the microbiome’s impact on stem cells, the role of antibiotics and probiotics in CRC, and the study of “omics” as pertains to the microbiome, which are beyond the scope of this review [52,58,59,60,61,62,63]. Instead, we will primarily focus on the role of the microbiome on inflammatory and select metabolic pathways in the context of colorectal cancer.

### 3.1. Direct Impact on Inflammatory Pathways

Several studies have identified a direct impact of pathogenic bacteria on colorectal inflammatory pathways. Many of these bacteria are also upregulated in CRC. For instance, *Fusobacterium* upregulates microRNA-21, which is associated with the activation of IL-10 and subsequent suppression of anti-tumor T-cell-mediated immunity [64]. *Fusobacterium* has also been shown to trigger tumor-associated macrophages through IL-6/STAT3/c-MYC signaling [65]. A few studies interestingly suggest that *Fusobacterium* may even promote the resistance of chemotherapeutic agents by CRC cells through the targeting of TLR4/NF-κB and MYD88 pathways, signifying that targeting pathogenic bacteria may lead to enhanced chemotherapeutic implications for CRC [66,67]. A recent study showed increased gene expression of proinflammatory cytokines such as TNF, IL-6, IFN-γ in *Fusobacterium nucleatum*-treated mice compared to controls [68]. Regarding other bacteria, *B. fragilis* toxin (BFT) was shown to trigger pro-carcinogenic inflammatory proteins including IL-17R, NF-κB, and STAT3 in colonic epithelial cells [69]. *Streptococcus gallolyticus* has been linked to increased RNA expression of NF-κB and IL-8 which aid in tumor formation [70]. *Enteroccocus faecalis* and *E. Coli* have been shown to stimulate COX-2 and PGE2 expression after infecting macrophages, which ultimately leads to a pro-tumorigenic environment [71].

### 3.2. Intestinal Barrier Disruption

The gut epithelium constitutes a diverse protective microenvironment, which contains commensal bacteria that produce secretory IgA, epithelial cells that help potentiate the activation of macrophages based on absorbed toxins, and immune cells in the lamina propria which secrete cytokines to help clear harmful pathogens [72,73]. All of these processes are intimately coordinated to maintain a healthy gut and to prevent the invasion of pathogens. Tight junctions are key components in maintaining this microenvironment by acting as intercellular gatekeepers, which prevent the entry of proinflammatory endotoxins, carcinogens and other toxic metabolites, such as genotoxins, across the epithelium [74]. Some studies have identified tight junction targets utilized by pathogenic bacteria to invade intestinal epithelium. *Fusobacterium nucleatum* has been shown to promote colorectal cancer by using FadA adhesin to bind E-cadherin and invade epithelial cells, with the subsequent activation of Wnt/β-catenin signaling pathway, one that promotes malignant cellular proliferation [75,76]. Enterotoxigenic *Bacteroides fragilis* increases intestinal permeability and induces a pro-carcinogenic state of persistent colitis by targeting E-cadherin [75]. Enterotoxigenic *E. Coli* also targets target tight-junction proteins such as occludin, claudin1, and ZO-1 through myosin light-chain kinase (MLCK)-myosin II regulatory light-chain (MLC20) pathways [77].

Obesity also plays a direct synergistic role, along with the gut microbiota, in altering intestinal barrier function [78]. Nagpal et al. present evidence that obesity-associated gut microbiome dysbiosis leads to derangements in intestinal cellular turnover homeostasis, as well as decreasing the expression of tight junction proteins [78]. Moreover, patients with higher BMIs have increased blood levels of Lipopolysaccharide (LPS), suggesting that obesity-induced barrier dysfunction may lead to the invasion of harmful bacterial toxins such as LPS [79]. One study showed that compared to patients with low levels of plasma LPS, those with higher levels were more likely to have colorectal adenomas [80]. In addition, patients with more pathologically advanced adenomas with villous histology were more likely to have higher endotoxin levels than those with tubular adenomas [80]. Collectively, these studies highlight the crosslinks between obesity, bacterial dysbiosis, intestinal barrier dysfunction and direct colorectal tumor formation.

### 3.3. Implications of the Microbiota on Genomics, Epigenetics and Metabolism

When the intestinal barrier is compromised, a plethora of pathogenic bacteria, toxins and oncometabolites are granted entry into intestinal epithelial cells. Many of these have the potential to cause genomic instability, a hallmark of carcinogenesis. Colibactin is one such alkylating genotoxin, produced by pks^+^ *E. coli*, and has been associated with the development of colorectal cancer in humans [81]. Moreover, a unique mutational signature caused by colibactin-producing *E. coli* was recently identified in human colon cancer genomes [82,83]. This exciting new evidence is the first of its kind to suggest the direct involvement of a bacterial toxin in human genome alteration, with subsequent development of colorectal cancer. The cytolethal distending toxin (CDT) is a protein exotoxin primarily produced by gram negative bacteria [84], which also exhibits cytotoxic effects [85]. One of the subtypes of CDT, CDTb, has been implicated in the development of single- and double-DNA-strand breaks [86]. In animal studies, CDT-producing bacteria have been shown to directly induce colorectal tumorigenesis [87,88]. Lastly, cytotoxic necrotizing factor 1 (CNF1) is a protein toxin produced by pathogenic *E. Coli,* which leads to multiple perturbations of the cell cycle, [89] including cell-cycle arrest and the stimulation of quiescent cells into proliferation [90]. Interestingly, Zhang et al. were able to demonstrate the phenomenon of reversible senescence, a mechanism suggested to provide a survival route for malignant cells, in human colorectal cancer cells [91]. Some bacteria are also able to produce reactive oxygen species (ROS), which can lead to DNA damage through oxidative processes [92]. For example, in the IL-10^−/−^ mouse model, *Enterococcus faecalis* has been shown to produce superoxide, which leads to a subsequent increase in 4-hydroxy-2-nonenal production by immune cells, a biochemical known to cause DNA instability [93]. *Peptostreptococcus. anaerobius*, a gram-positive anaerobe also correlated with CRC, has been shown to support cholesterol formation and cellular dysplasia via TLR2 and TLR4, which leads to the accumulation of reactive oxygen species [94]. Furthermore, hydrogen sulfide production by enteric bacteria such as *Fusobacterium nucleatum* and *E. coli* has also been linked to ROS-mediated DNA damage [95].

Another emerging topic of discussion in CRC biology is the role of the microbiome in epigenetics. *Fusobacterium nucleatum* has been associated with epigenetic alterations via genomic hypermutation, correlation with the CIMP phenotype, wild-type TP53, hMLH1 methylation, and CHD7/8 mutation [96,97]. *Bacteroides fragilis* toxin leads to changes in DNA methylation, which ultimately contribute to the formation of CRC [98] (see Figure 2).

In contrast, epigenetic changes by bacteria could potentially have anti-neoplastic effects. It has been shown that bacterial metabolites such as SCFAs (butyrate, acetate) can lead to epigenetic changes within the intestinal epithelium [99] via the inhibition of histone deacetylase, which causes changes in the cancer cell expression of cell regulatory proteins to an anti-neoplastic phenotype [100]. In addition, butyrate has been shown to stimulate T_reg_ induction by histone acetylation of CD4^+^ T cells, a pathway which is involved in anti-tumor activity [101]. These effects are further enhanced by the increased amounts of fiber in the intestines, which highlights the intricate crosstalk between the microbiome and diet on the colorectal cancer milieu [102,103].

### 3.4. Protective Implications of Microbiota and Future Directions

The microbiota also exhibit protective effects against colorectal neoplasm formation. In addition to inducing epigenetic changes, butyrate also interacts with epithelial cells to increase IL-18, which helps stimulate tumor-fighting TH1 cells [104]. Butyrate also induces IL-10 and Aldh1a, which trigger the differentiation of naïve T cells into T_reg_ cells and the suppression of Th17 cells, which have been implicated in tumorigenesis [105]. Furthermore, in a study of colon microbiota of patients with CRC, there was an observed decrease in butyrate-producing bacteria, which belonged to the phylum *Firmicutes,* suggesting that SCFAs may play an essential role in preventing the development of the tumor microenvironment [106]. In vivo studies showed that isothiocyanates, microbial metabolites that have been identified in many intestinal bacteria such as *E Coli*, *Enteroccocus faecalis*, and *Peptostreptococcus*, prevent tumorigenesis via modifications of miRNA, histones and DNA methylation [107,108]. Amuc_1434, a recombinant protein derived from *Akkermansia muciniphila*, causes the inhibition of colorectal cancer cell lines via *E. Coli* prokaryote cell system [109]. More specifically, Amuc_1434 leads to the degradation of Muc2, a gene that leads to the production of mucin 2, an intestinal protein previously shown to be significantly increased in colorectal cancer [109]. Nontoxigenic *B. fragilis* has been shown to play a protective role against colitis-associated CRC in an animal study via TLR2 signaling [69]. In addition, the outer membrane vesicles of nontoxigenic *B. fragilis* suppress proinflammatory receptors and cytokines such as TLR2 and IFN-γ, while increasing the expression of anti-inflammatory cytokines IL-4 and IL-10 in Caco-2 colon cells [69]. This bidirectional effect of the microbiome on pro-carcinogenic and anti-carcinogenic pathways, as well as its intimate influence on the genomic, epigenetic, and metabolic levels of the intestinal ecosystem, suggests the need for a refined approach to our study of the microbiome and colon cancer. This is further aided by the advent of “omics” such as metabolomics, metagenomics, and metaproteomics.

Given that the microbiota plays an important role in the inflammatory pathogenesis of CRC, the genetic–environmental contributions to the gut microbial composition is paramount to our understanding of CRC. In a recently published study, the microbiome of children who were unrelated genetically but living in a similar environment was compared to genetically related children living in different environments [110]. Using generalized dissimilarity modeling, it was found that the quantity of a specific microbial taxa was explained by host genetic similarity, whereas the species composition was dependent on environmental factors [110]. This is especially important in the context of CRC, given that certain bacterial species are more likely to be associated with CRC development. For example, *Escherichia coli* and *Bacteroides fragilis* were found in higher quantities in colorectal mucosal tissues from patients with familial adenomatous polyposis, caused by germline APC mutations [111]. A genome-wide association study for CRC showed that the C2CD2-gene single-nucleotide polymorphism was associated with a higher incidence of colorectal adenomas [112]. In a Japanese study, the bacterial family *Erysipelotrichaceae*, which is known to produce intestinal inflammation, was found to be associated with the C2CD2-gene single-nucleotide polymorphism, suggesting that the C2CD2 gene leads to CRC through the microbiota [113]. By identifying the genetic variants that lead to microbial-induced intestinal inflammation and predisposition to CRC formation, we create a unique opportunity for risk stratification and prevention of CRC. In addition, germline genetic variations may play a role in the treatment of CRC. In a study consisting of 361 patients, pathogenic germline variants were found in 15.5% of patients with CRC [114]. A total of 9.4% of patients had findings that would not have been detected by practice guidelines or CRC-specific gene testing [114]. Furthermore, 11% of patients had modifications in their treatment based on the pathogenic germline variants [114].

Another concept that is paramount to our understanding of genetic–environmental implications on the microbiome is molecular pathological epidemiology (MPE). MPE studies are largely based on the principle that human disease is largely heterogeneous due to both intrinsic factors (germline genetic variations, sex) and extrinsic factors (lifestyle, acquired alterations in microbiota) [115]. Given this heterogeneity, MPE is able to subclassify patients with a disease into more homogenous groups using molecular pathological markers [115]. Using this approach to CRC, we can create more etiologic links between germline genetic variations, and environmental factors with specific subtypes of CRC based on potential immunologic or microbial profiles [111]. This has led to recent subfields of MPE including immuno-MPE and microbial-MPE [115].

As we have discussed, the immune system plays a considerable role in the inflammatory pathogenesis of CRC and is affected by risk factors such as smoking, obesity and diet, and diabetes. Immuno-MPE studies in relation to CRC have demonstrated insight into immunomodulators for CRC immunoprevention [111]. For example, one study showed that the risk reduction of CRC with high FOXP3^+^ T cell infiltrates was higher when associated with an increased intake of marine ω-3 Polyunsaturated fatty acid (ω-3 PUFA), suggesting that marine ω-3 PUFA may create antitumor effects by inhibiting regulatory T cell function [116]. Furthermore, MPE studies have helped link intestinal immunogenicity to the microbiome in the context of CRC. For example, *Fusobacterium nucleatum* in colorectal tumors was associated with lower levels of tumor-infiltrating lymphocytes (TIL) in MSI-high CRC [117]. Conversely, it was associated with high-level TIL in non-MSI-high carcinoma, suggesting that bacteria may play a direct role in the tumor-immune microenvironment phenotype [117]. In addition, MPE studies have demonstrated that a diet rich in processed and red meat is associated with a higher risk of *Fusobacterium-nucleatum*-positive proximal colon cancer, but not with a higher risk of *Fusobacterium-nucleatum*-negative proximal cancer [118]. On the contrary, a diet rich in whole grains and fiber has been associated with a lower risk of *Fusobacterium-nucleatum*-associated CRC but not with a lower risk of *non-fusobacterium-nucleatum*-associated CRC [119].

Although MPE studies serve as great tool to help with the individualized early detection and prognostication of CRC, they still present some challenges. MPE analyses tend to be limited to patients who are able to provide biospecimens, which tends to limit the study sample size [120]. Furthermore, biospecimen collection necessitates a rigorous standardization process of collecting and processing individual samples [111]. Since MPE studies often cover multiple disciplines, multidisciplinary research teams, which include fields such as molecular pathology, immunology and microbiology, are required [111]. Finally, given that MPE studies explore heterogeneous diseases, false-positive findings may arise, given the inevitability of multiple hypothesis testing [115].

In summary, research on genetic–environmental contributions, germline genetic variations, and MPE studies in CRC only reiterates the new, exciting era of medicine we have entered, which focuses on personalized precision medicine. These modalities of studying CRC provide unique opportunities for the risk stratification of patients as well as enhanced early detection and prevention of this deadly disease.

## 4. Mycobiome

Within the last decade, there has been an exponential increase in research regarding the role of commensal fungi in human disease. The “mycobiome” entails the plethora of fungi that are found on different body surfaces such as the lungs, oral cavity and small and large intestinal tissues. The advent of the Human Microbiome Project revealed that the four most abundant genera in the human intestines are *Saccharomyces, Malassezia, Candida*, and *Cyberlindnera* [121]. Interestingly, unlike the gut bacterial microbiome, the gut mycobiome appears to be much more variable and transient over time between different humans [121,122]. Recent studies have suggested this variability may be partly due to changes in diet. In a recent animal model study, Mims et al. show that mice fed with processed diets have significantly altered fungal communities compared to mice fed with a less processed diet [123]. Furthermore, an increase in the fungal genera *Thermomyces* and a decrease in *Saccharomyces* was associated with metabolic disturbances such as increased triglyceride concentrations and the hepatic deposition of lipids [123]. Another metabolic and immunologic link between fungi and the gut involves Dectin-1 (CLEC7A), a human fungal-recognition receptor expressed by immune and adipose cells [124,125]. Not only has Dectin 1 expression been shown to be increased in the fat cells of obese individuals, but its antagonism is also associated with improved glucose homeostasis and decreased CD11c^+^ AT macrophages in chow- and HFD-fed MyD88 knockout mice [125]. This suggests a unique crosstalk between the gut mycobiome, and metabolic and proinflammatory pathways in the gut.

Several studies have established a direct link between the gut mycobiome and colorectal adenomas/cancer. Luan et al. showed a difference in overall mycobiome between adenomas and adjacent healthy biopsy samples through operational taxonomic unit (OTU)-level analysis [126]. Specifically, the genera *Candida* and *Phoma* were found to be crucial in the formation of adenomas [126]. One clinical study, comprising 131 subjects, 74 of whom had colorectal cancer and 29 who had adenomatous polyps, revealed fungal dysbiosis, characterized by an increased *Ascomycota/Basidiomycota* ratio, in patients with colon cancer [127]. In addition, there were increased proportions of *Trichosporon* and *Malassezia* coinciding with the progression of cancer (i.e., early-stage vs. late-stage cancer) [127]. Another study identified 14 fungal biomarkers distinguishing 184 CRC patients in comparison to 204 control subjects [128]. Interestingly, they revealed an increased *Basidomycota/Ascomycota* ratio in colorectal cancer vs. healthy subjects; however, Malasseziomycetes were increased in CRC, while *Saccharomycetes* and *Pneumocystidomycetes* were decreased [128].

The potential mechanisms by which the mycobiome is directly involved with the development of colorectal cancer can be organized into two broad categories: activation of the immune response and toxic fungal metabolites leading to cellular/DNA damage.

The immune response primarily recognizes fungal pathogens through the binding of pathogen-associated molecular patterns (PAMPs) via pattern-recognition receptors (PRRs) [129]. This binding leads to the activation of signaling cascades such as SYK-CARD9, which ultimately leads to the production of pro-inflammatory cytokines such as IL-1β, IL-6, IL-12, IL-23, and IFN-γ [129]. Caspase recruitment domain-containing protein 9 (CARD9) is an intracellular adaptor protein predominantly expressed by myeloid cells [130]. CARD9 has been shown to be directly involved in the immune response via neutrophil recruitment, macrophage polarization and the differentiation of myeloid derived suppressor cells (MDSC). The expression of CARD9 in the large intestine has been found to be significantly higher in CRC patients with a low fungal burden compared to those with relatively high fungal burdens [131,132]. CARD9 deficiency also leads to the intestinal growth of Candida Tropicalis, which is directly linked to the differentiation of MDSCs and subsequent suppression of T cells, a pro-tumorigenic process [132]. Interestingly, some evidence suggests that CARD9 may play a protective role against colitis and CRC through the SYK-CARD9–Il-18 axis [130] (see Figure 3).

Recently, Zhu et al. were able to demonstrate that increased amounts of intestinal Candida albicans directly lead to glycolysis in macrophages and subsequent interleukin-7 (IL-7) secretion [133]. IL-7 leads to IL-22 production in RORγt^+^ innate lymphoid cells via aryl hydrocarbon receptor and STAT3 activation [133]. IL-22, which is primarily produced by CD4^+^ cells, has not only been shown to contribute to gastrointestinal inflammation [134], but also to stimulate colorectal tumor growth via the activation of STAT3 transcription factor and H3K79 methyltransferase [135].

DNA damage and genomic instability have long been deemed fundamental components in the narrative of carcinogenesis. Fungi have the potential to contribute to this narrative via mycotoxins. Mycotoxins are low-molecular-weight compounds, mainly produced mainly by fungi, which have genotoxic effects that lead to carcinogenesis [136]. The most notorious mycotoxin is aflatoxin, which is well known for its contributions to hepatocellular carcinoma. In addition to aflatoxin, many other toxins have been identified as having carcinogenic activity such as fumonisins, ochratoxins, trichothecenes, and zearalenone [136,137]. Kwan Lo et al. recently showed that low-dose zearalenone stimulated colon cancer cells in vitro through a G-protein-coupled estrogenic receptor [138]. Furthermore, zearalenone causes decreased expression of tight junction proteins with subsequent gut barrier dysfunction as well as gut bacterial dysbiosis [139]. In addition, mycotoxins have been well-documented in the Fusarium genus [140], which is prevalent in patients with colon cancer [141] (see Figure 4).

*Candida albicans* also promote carcinogenesis through the production of nitrosamine and acetaldehyde [142]. Acetaldehyde not only leads to DNA damage and oxidative stress, but also to tight junction disruption [143]. Candida albicans also secrete Candidalysin, a peptide toxin which leads to chromosomal translocation [144]. Mycotoxins, acetaldehyde and candidalysin are biochemical mediators that were recently identified to potentially play a role in colon carcinogenesis. Since they are involved on multiple fronts, including DNA damage, bacterial dysbiosis and gut barrier disruption, further investigations are necessary in order to produce a more dynamic and holistic understanding of their role in colorectal cancer biology.

In addition to the aforementioned mechanisms by which fungi may contribute to the development of colon cancer, fungi also have a direct interaction with gut bacteria. Fungal–bacterial interactions have been shown to synergistically lead to pathogenesis, as well as cellular disruption [144]. Fungi directly aid in the formation of bacterial biofilms, leading to intestinal inflammation [122,145]. One recent study showed that in germ-free mice who became colonized with defined species of fungi, bacteria, or both entities, fungal colonization was shown to produce major shifts in the intestinal bacterial ecology [145]. It also led to independent effects on innate and adaptive immunity [146]. As both gut bacteria and fungi independently have the potential to contribute to the inflammatory pathogenesis of CRC, the close interaction between these two entities provides an exciting potential to produce unique early diagnostic signatures based on gut bacterial and fungal profiles. These discoveries further highlight the need to approach our study of colorectal cancer through a multidisciplinary lens, one that includes inflammation, the microbiome, and the mycobiome.

## 5. The Role of Exosomes

Extracellular vesicles (EVs), are membrane-contained vesicles released by all types of cells, which can originate from intracellular endocytic pathways or from the cell plasma membrane [147]. Initially, EVs were considered a means of releasing intracellular waste including proteins; however, they were later found to have precious cargo including lipids, DNA, protein, RNA, and microRNA, which play a crucial role in intercellular communication [147]. EVs can be found in all bodily fluids including urine, plasma, cerebrospinal fluid, and amniotic fluid [148]. Several centrifugation techniques have been developed to successfully extract these microvesicles for laboratory study [149]. Exosomes, a specific subtype of EVs, range from 30 to 120 nm and were previously characterized as containing 9769 proteins, 1116 lipids, 3408 mRNAs, and 2838 miRNAs [150]. Although the study of all of these components provides insight into the realm of intercellular communication, microRNAs and long-coding RNAs have sparked particular interest given their intimate involvement with post-transcriptional gene expression and signal transduction pathways. Exosomes have been shown to have key functions in biological processes such as immune modulation, angiogenesis, coagulation, tissue regeneration, and neoplasm formation [150]. Here, we primarily focus our discussion on the role of exosomes in inflammatory and pro-carcinogenic pathways as they relate to colorectal cancer.

### 5.1. Exosomes and Intestinal Inflammation

The role of exosomes in intestinal inflammatory pathways has been extensively studied in inflammatory bowel disease patients. For example, exosomes extracted from macrophages and neutrophils from IBD patients and absorbed by intestinal epithelial cells have been shown to lead to the epithelial release of IL-8 and subsequent macrophage recruitment in vitro [151]. Elevated levels of IL-6, IL-8, IL-10 and TNF-α were found in exosomes extracted from the intestinal lumen of IBD patients when compared to controls [152]. Furthermore, exosomes extracted from the intestines of IBD patients were found to contain elevated levels of myeloperoxidase, an enzyme that creates reactive oxygen species (ROS), which lead to inflammation [153]. In addition, the intestinal barrier integrity has also been shown to be modulated by miRNAs released by exosomes [154]. One of these mi-RNAs, miR-21, is not only upregulated in patients with IBD, but also consistently causes the downregulation of E-cadherin and ZO-1 proteins in vitro and in vivo [155,156]. MiR-223 is also upregulated in IBD patients when compared to healthy subjects. Moreover, it is a contributor to the IL-23 pathway, a major pathway in IBD pathobiology, through suppression of the tight-junction protein Claudin-8 [157,158].

Wei et al. were recently able to show a link between obesity, exosomes and colitis. Mice fed a high-fat diet were found to have altered miRNA profiles of visceral adipose tissue exosomes, causing a change in the exosome phenotype from anti-inflammatory to pro-inflammatory. Furthermore, these exosomes were injected into mice treated with dextran sodium sulfate, a chemical used to mimic colitis in mice, and exosomes were found to have migrated to the lamina propria of the intestines, with subsequent M1 macrophage polarization and worsening mucosal damage via miR-155. This effect was significantly inhibited by the administration of the miR-155 inhibitor, signifying a potential therapeutic implication for exosomes in the study of colitis [159].

Fascinatingly, exosomal-mediated intestinal inflammation is not just limited to eukaryotic cells. The microbiome has also been found to be a major contributor to this paradigm. One study showed that exosomes derived from intestinal cells and macrophages infected with adherent-Invasive E. coli led to activated NFkB, mitogen-activated protein kinases p38, and increased secretion of proinflammatory cytokines [160]. Additionally, exosomes from infected cells also increased bacterial replication compared to exosomes from uninfected cells [160]. This finding suggests the use of exosomes as transport vehicles by bacteria to disseminate their virulence factors. This is further supported by a study that identified cytotoxin-associated gene A (CagA), the major virulence factor secreted by *H. pylori*, in the exosomes of gastric cells, expressing the CagA gene [161]. Prostaglandin E2 and the immune chemokine CCL20 were identified in the exosomes of *Bacteroides fragilis*, one of the bacteria that were previously identified to be associated with colon cancer [162]. Prostaglandin E2 has been associated with the suppression of Tregs as well as B-lymphocyte differentiation [163]. In a recent study, exosomes derived from Fusobacterium nucleatum were found to promote the metastasis of cancer cells via miR-1246/92b-3p/27a-3p and CXCL16 [164]. Another study found evidence of intestinal barrier disruption through the targeting the RIPK1-mediated cell death pathway by exosomes derived from *Fusobacterium nucleatum* [165].

Exosomes provide a modality that not only augments our understanding of the roles of obesity and the microbiome in CRC, but also provides a means to target some of the proinflammatory environments created by these conditions. For example, an animal model study showed evidence of the suppression of intestinal dysbacteriosis-induced tumorigenesis of colorectal cancer through the injection of an exosome secretion inhibitor, neticonazole [166]. The CRC cell line SW480 was inoculated as a xenograft tumor in an intestinal dysbacteriosis mouse model, and tumor growth was monitored for 15 days [166]. Neticonazole was subsequently injected into these mice, and decreased tumor growth was observed, as well as a significant improvement in the survival of mice treated with neticonazole [166].

### 5.2. Exosomes and Colorectal Cancer

The direct impact of exosomes on CRC can be categorized into three broad areas: modulation of the immune system, role in cell proliferation, and role in metastasis.

Exosomes from CRC cells have been shown to contain anti-CD8^+^ apoptotic molecules, such as TNF-related apoptosis-inducing ligands such as the Fas ligand (FasL) [167,168]. One study further analyzed the effect of CRC exosomes on CD8^+^ functionality from high- versus low-BMI patients [169]. Although there was no statistical significance found in the T cell functional assays between high- and low-BMI groups, there was an abundance of CD8^+^ cells and a reduction in anti-tumor NK cells in the highest-BMI group when compared to the low-BMI group, further highlighting the pro-inflammatory role of obesity [170]. The serum exosomal long-coding RNA CRNDE-h level was found to be positively correlated with the proportion of Th17 cells, a previously identified contributor to the development of malignant tumors [169]. MiR-203 was found to promote M2- tumor-associated macrophages (TAMs) when internalized from exosomes of CRC cells [171]. miR-1246, discovered in colon cancer cell exosomes, was identified as a key mediator in the transformation of macrophages to the tumor-supporting phenotype [172]. The expression of VEGF, Wnt5A and IL-1β were also found to be upregulated by CRC exosomes, subsequently leading to TAM differentiation [173]. Furthermore, TAMs-derived exosomal miR- 155-5p was recently found to cause an “immune escape” phenomenon, where malignant cells actively evade the immune system through the indirect downregulation of IL-6 [174].

Regarding cellular proliferation and metastasis, several studies have identified key exosomal players, which aid these phenomena in colon cancer development. Exosomal WNT1 was shown to lead to the increased proliferation and migration of CRC cells [175]. Another study revealed evidence that exosomal miR-30a and miR-222, derived from colon cancer mesenchymal stem cells, promoted the tumorigenicity of colon cancer [176]. MiR-424-5p was also found to promote the proliferation and metastasis of colorectal cancer by directly inhibiting SCN4B, a tumor-suppression gene [177]. The elevated expression of serum exosomal miR-203 is associated with distant metastasis and serves as an independent poor prognostic factor of CRC [171]. Moreover, exosomes from CRC cells were shown to induce a malignant phenotype when injected into normal colonic cells in vitro [178]. Interestingly, the reverse was also true, where exosomes from normal colon cells reversed the malignant phenotype of colon cancer cells, suggesting that exosomal cargo may contain a key putative therapeutic potential [178].

In summary, exosomes have greatly advanced our knowledge of the inflammatory pathways, intercellular communication and direct pro-carcinogenic and pro-metastatic mechanisms involved in colorectal cancer (see Figure 5).

Importantly, they also have great potential as potential biomarkers and vehicles of therapeutic intervention. For example, over 20 miRNAs and long-coding RNAs have been directly and exclusively linked to colon cancer diagnosis [179]. In addition, dozens more have been directly linked to metastasis, potential chemotherapy resistance and neoplastic recurrence [179]. Recently, there has been encouraging data regarding the effects of anti-inflammatory and anti-neoplastic-related miRNAs on colon cancer cells, as well as the effects of pro-inflammatory/pro-neoplastic miRNA inhibitors on the tumor microenvironment [180,181,182,183,184,185]. Although further studies are necessary before the therapeutic value of exosomes is moved from the laboratory to the bedside, their diagnostic potential and their pertinence in uncovering the inflammatory pathways involved in CRC is invaluable.

## 6. Conclusions

In recent years, our understanding of the pathobiology of colorectal cancer has greatly evolved. In addition to the adenoma -to-carcinoma sequence, which delineates oncogenic mutations as a fundamental component in carcinogenesis, we now have a strong body of evidence regarding the inflammatory and metabolic pathways as paramount players in the development of CRC. The study of CRC from an inflammatory perspective not only helps to uncover some of the underlying mechanisms of this deadly disease, it also creates a common denominator for the different risk factors and helps to pinpoint specific biologic targets for more accurate diagnostic and therapeutic interventions. For example, in our discussion regarding obesity and diet, smoking and diabetes, the microbiome and the mycobiome, we were able to demonstrate how these entities share similar implications on the proinflammatory pathways while affecting similar cytokines and transcription factors such as IL-6, IL-1, IL-17, STAT3, NFkB (see Figure 6).

Furthermore, approaching CRC from an inflammatory perspective makes it simple to incorporate other disciplines such as exosomes and the role they play in the pathogenesis and potential diagnosis of, and therapy for, CRC. It also aids in creating new questions that need to be answered. What is the effect of exosomes from fungi on the intestinal neoplastic milieu? How soon can exosomes have bedside therapeutic potential? By targeting certain inflammatory pathways caused by obesity, can the microbiome and mycobiome create a realistic, clinical target to prevent CRC tumor formation? Furthermore, in addition to obesity, what are the pro-inflammatory and pro-carcinogenic pathways caused by other established CRC factors such as diabetes and smoking? What similarities do these pathways share with those of obesity, the microbiome and the mycobiome? What exosomal miRNAs are involved in these processes? What roles do negatively associated risk factors of CRC such as exercise and dairy product consumption have on inflammation? What can molecular pathological epidemiology studies uncover with respect to exo-somes, CRC and the microbiome? Through an inflammatory and metabolic lens, and with exciting new tools such as germline genetic variations analysis and molecular pathological epidemiological studies, we comprehensively answer these questions with personalized precision. Additionally, our knowledge of CRC becomes more refined and organized, allowing us to combine different associations and risk factors of CRC, instead of viewing them as mutually exclusive entities.

## Figures and Tables

**Figure 1 diseases-09-00079-f001:**
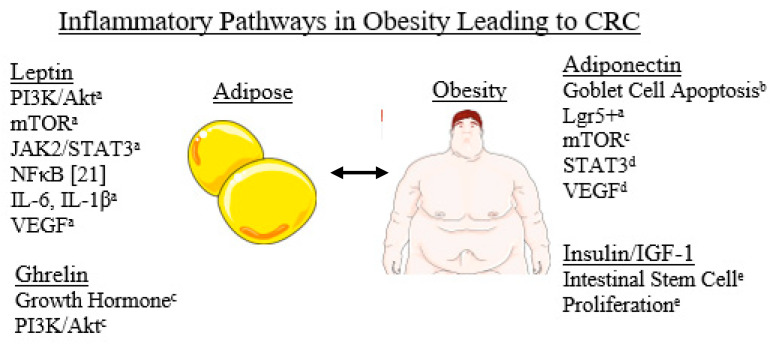
Hormones implicated in obesity, with the respective inflammatory cytokines and transcription factors they affect. ^a^: Chang et al. 2020, ^b^: Saxena et al. 2013, ^c^: Moodi et al. 2021, ^d^: Moon et al. 2012, ^e^: Pourvali et al. 2021.

**Figure 2 diseases-09-00079-f002:**
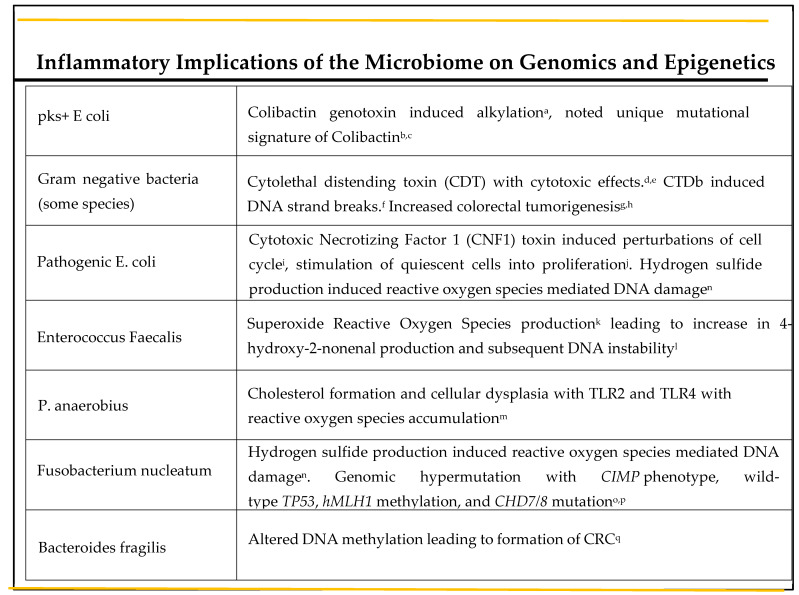
A depiction of how bacteria can lead to genetic and epigenetic changes of intestinal epithelium via DNA methylation, genotoxins and reactive oxygen species. ^a^-Bonnet et al. 2014, ^b^-Arthur et al. 2020, ^c^-Dziubańska-Kuibab et al. 2020, ^d^-Gargi et al. 2012, ^e^-Guerra et al. 2009, ^f^-Fahrer et al. 2014, ^g^-Buc et al. 2013, ^h^-He et al. 2019, ^i^-El-Aouar Filho et al. 2017, ^j^-Giamoi-Miraglia et al. 2007, ^k^-Srinivas et al. 2019, ^l^-Gunasekera et al. 2020, ^m^- Tsoi et al. 2017, ^n^-Barrett et al. 2020, ^o^-Inamura et al. 2018, ^p^-Koi et al. 2018, ^q^-Allen et al. 2019.

**Figure 3 diseases-09-00079-f003:**
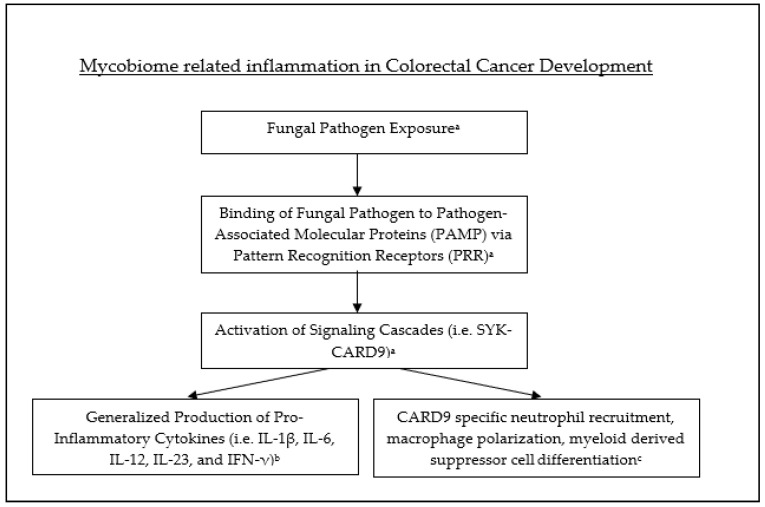
Fungal pathogen exposure leads to binding with PAMP, subsequent activation of signaling cascades and production of inflammatory cytokines as well as recruitment and activation of inflammatory cells. ^a^-Hatinguias et al. ^b^-Malik et al. ^c^-Bertin et al.

**Figure 4 diseases-09-00079-f004:**
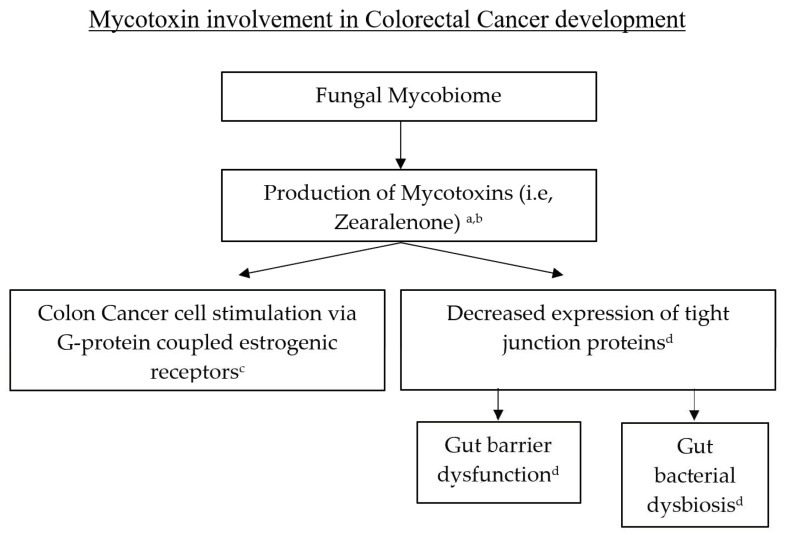
Mycotoxins can lead to CRC development through G-protein coupled estrogenic receptors or via gut barrier disruption and gut bacterial dysbiosis. ^a^-Omotayo et al., ^b^-Buszewska-Forajita et al., ^c^-Lo et al., ^d^-Zhang et al.

**Figure 5 diseases-09-00079-f005:**
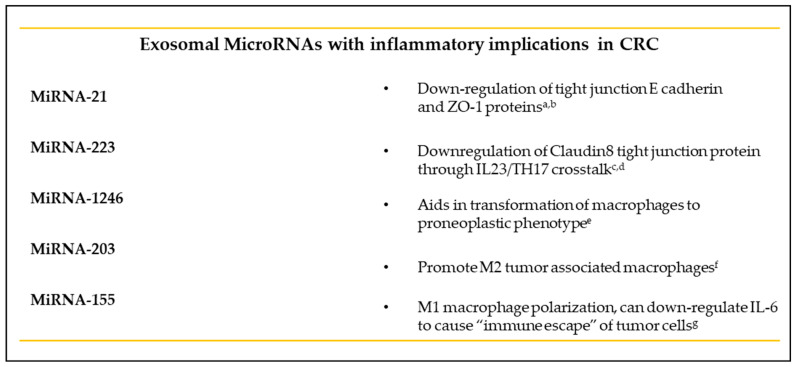
Exosomal microRNAs which have implications in CRC development and the various pathways and inflammatory cells which they affect. ^a^-Zhang L et al. 2015, ^b^-Zhang et al. 2018, ^c^-Yuan et al. 2018, ^d^-Wang et al. 2016, ^f^-Takano et al. 2011, ^g^-Ma et al. 2021.

**Figure 6 diseases-09-00079-f006:**
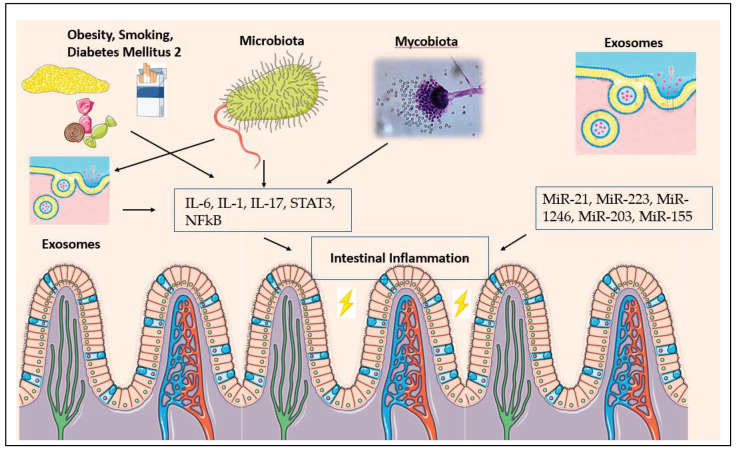
Crosslinks between obesity, cigarette smoking, diabetes mellitus tye 2, microbiota, mycobiota, and exosomes and their collective impact on creating intestinal inflammation.

## Data Availability

Not applicable.

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
