# Peer review of "Exploring the Inflammatory Pathogenesis of Colorectal Cancer"

_diseases, 2021, doi:10.3390/diseases9040079_

Round 1

Reviewer 1 Report

This is an interesting paper that reviews the role of inflammation

in colorectal cancer (CRC). The authors discussed it from the obesity, the microbiome, and mycobiome aspects. I have some comments.  

  1. Page 1. “in both men and women”. Why do you not use “in people”?
  2. The captions of Figures 1-5 should be rephrased to illustrate these figures instead of only listing the cited references.
  3. The fonts of the citation in the second box of Figure 4 are not consistent.
  4. In Section 2, obesity and diet in Colorectal Carcinoma were discussed. However, you mainly discussed obesity from the processed and red meat consumption viewpoints. However, many vegetarians also have the obesity problem. Therefore, I recommend adding more content in discussing the relationship between obesity and CRC.

Author Response

We thank reviewer 1 for their insightful comments and critiques. We have incorporated their suggestions and revised the manuscript accordingly. We believe that their input has substantially improved our manuscript. Below is a point-by-point response and changes are highlighted in red color in the revised redline version of the manuscript.

Q

Page 1. “in both men and women”. Why do you not use “in people”?

A

Thank you for this suggestion. We have edited the text to mention “people”.

Q.

The captions of Figures 1-5 should be rephrased to illustrate these figures instead of only listing the cited references.

A

We have rewritten the captions for all of the figures to illustrate what the figures entail.

Q

The fonts of the citation in the second box of Figure 4 are not consistent.

A

We have updated the fonts in figure 4 as well as in the rest of the figures to maintain a uniform font.

Q.

In Section 2, obesity and diet in Colorectal Carcinoma were discussed. However, you mainly discussed obesity from the processed and red meat consumption viewpoints. However, many vegetarians also have the obesity problem. Therefore, I recommend adding more content in discussing the relationship between obesity and CRC.

A

While the aforementioned sections regard the role of short chain fatty acids as well as the implications of diets with red meat excess, attention has also been directed towards all-cause obesity in the pathogenesis of colorectal carcinoma. As mentioned in obese individuals exists a higher prevalence of colorectal cancer formation, attention has been directed towards the fat mass and obesity (FTO) pathway (Roslan et al 2019). In this genotype, a FTO protein is linked to fat mass in humans and is predominantly located in the hypothalamus with control over satiety, and affects the intake of food overall as opposed to energy expenditure (Roslan et al 2019). Interestingly, this purports the notion that regardless of dietary composition, genetic changes influencing excess food intake may play a role in obesity and therefore colorectal pathogenesis.

While genetic factors have been studied in carcinogenesis, these factors are also augmented dependent on the region of habitation. For example, it has been shown that Japanese immigrants moving to Hawaii will acquire the same risk of colorectal cancer, attributed to diets rich in the aforementioned food groups in lacking in non-refined grains and low in fruits and vegetables (Song et al 2019). High fat diets have demonstrated higher development of colorectal carcinoma and in part this is due to a lack of protective elements found in fruits and vegetables (Roslan et al 2019). Phytochemicals are micronutrients synthesized from plants and play a role in the anti-inflammatory and antineoplastic effects to prohibit carcinogenesis (O’Keefe et al 2016). While certain food groups have demonstrated higher carcinogenesis, the lack of food groups such as fruits and vegetables in sum with genetic factors leading to obesity create the notion that with and without diet, obesity plays a role in increase carcinogenesis.  

References:

O’Keefe, S. J. D. (2016). Diet, microorganisms and their metabolites, and colon cancer. In Nature Reviews Gastroenterology and Hepatology (Vol. 13, Issue 12). https://doi.org/10.1038/nrgastro.2016.165

Roslan, N. H., Makpol, S., & Mohd Yusof, Y. A. (2019). A review on dietary intervention in obesity associated colon cancer. In Asian Pacific Journal of Cancer Prevention (Vol. 20, Issue 5). https://doi.org/10.31557/APJCP.2019.20.5.1309

Song, M., & Chan, A. T. (2019). Environmental Factors, Gut Microbiota, and Colorectal Cancer Prevention. In Clinical Gastroenterology and Hepatology (Vol. 17, Issue 2). https://doi.org/10.1016/j.cgh.2018.07.012

Reviewer 2 Report

The authors wrote a quite interesting review on inflammation and colo-rectal cancer. This is generally of interest.

Following things should be addressed and the authors should improve the paper.

They put “obesity, microbiome, mycobiome, and exosomes” in the title. This should be revised. These 4 things are not parallel concepts. Obesity is a morbidity. Mycobiome is a part of microbiome. Exosomes are biological components and are included in extracellular vesicles. These should not be in parallel. What if anyone put an article title “Inflammation, colorectum, cancer, obesity, microbiome, mycobiome, exosomes”. This just lists words. That’s not a good title. I hope the authors understand.  

Prevention should be the top topics. It is currently hidden in the middle. The authors should discuss more prevention and early detection as main topics since they are important in reducing the cancer burden much more effectively than treatment.

There are many environmental, dietary, and lifestyle factors that influence the microbiome (in both intestine and other tissue), inflammation, immune system, pathogenic mechanisms. The authors should discuss factors other than diet too, eg, smoking, alcohol, obesity, diabetes, bowel habits, etc.

There are also influences of germline genetic variations on both immune system and microbiota. Gene-by-environment interactions should be discussed.

In these lines, research on dietary / lifestyle factors, microbiome, immunity, and personalized molecular biomarkers in tumor is needed for prevention and treatment research. The authors should discuss molecular pathological epidemiology research that can investigate those factors in relation to microbiome, molecular pathologies, immunity, inflammation, and clinical outcomes. Molecular pathological epidemiology research can be a promising direction. Strengths and challenges of molecular pathological epidemiology (in J Pathol 2019, Ann Rev Pathol 2019, Curr Colorectal Cancer Rep 2017, etc.) should be discussed.

Author Response

We thank reviewer 2 for their insightful comments and critiques. We have incorporated their suggestions and revised the manuscript accordingly. We believe that their input has substantially improved our manuscript. Below is a point-by-point response and changes are highlighted in red color in the revised redline version of the manuscript.

Q.

They put “obesity, microbiome, mycobiome, and exosomes” in the title. This should be revised. These 4 things are not parallel concepts. Obesity is a morbidity. Mycobiome is a part of microbiome. Exosomes are biological components and are included in extracellular vesicles. These should not be in parallel. What if anyone put an article title “Inflammation, colorectum, cancer, obesity, microbiome, mycobiome, exosomes”. This just lists words. That’s not a good title. I hope the authors understand.

A

Thank you for this insightful remark. We have revised the title.

Q

Prevention should be the top topics. It is currently hidden in the middle. The authors should discuss more prevention and early detection as main topics since they are important in reducing the cancer burden much more effectively than treatment.

A

Thank you for this advice. We have updated the abstract, introduction, obesity and diet, microbiome, mycobiome and conclusion to reflect these changes. We have also elaborated on early detection and prevention of CRC through discussion of germline genetic variations and molecular epidemiological studies. Below we show the changes we made in their respective sections (not including discussion on germline genetic variations and molecular pathologic epidemiology).

Obesity and, Smoking, Diabetes

Epidemiological studies have shown an increased risk of colon cancer in people with high levels of leptin (Stattin et al). Other studies highlight an important interaction between leptin and soluble leptin receptor in the development of CRC (Aleksandra et al). Although more studies are needed to confirm specific mechanisms by which leptin is implicated in CRC development in obese individuals, these studies suggest potential novel approaches for refined risk stratification, and early diagnosis and prevention of CRC.

References

Aleksandrova, K., Boeing, H., Jenab, M., Bueno-de-Mesquita, H. B., Jansen, E., van Duijnhoven, F. J. B., Rinaldi, S., Fedirko, V., Romieu, I., Riboli, E., Gunter, M. J., Westphal, S., Overvad, K., Tjnøneland, A., Halkjræ, J., Racine, A., Boutron-Ruault, M. C., Clavel-Chapelon, F., Kaaks, R., … Pischon, T. (2012). Leptin and soluble leptin receptor in risk of colorectal cancer in the European prospective investigation into cancer and nutrition cohort. Cancer Research, 72(20). https://doi.org/10.1158/0008-5472.CAN-12-0465

Stattin, P., Lukanova, A., Biessy, C., Söderberg, S., Palmqvist, R., Kaaks, R., Olsson, T., & Jellum, E. (2004). Obesity and colon cancer: Does leptin provide a link? International Journal of Cancer, 109(1). https://doi.org/10.1002/ijc.11668

Stattin, P., Palmqvist, R., Söderberg, S., Biessy, C., Ardnor, B., Hallmans, G., Kaaks, R., & Olsson, T. (2003). Plasma leptin and colorectal cancer risk: a prospective study in Northern Sweden. Oncology Reports, 10(6). https://doi.org/10.3892/or.10.6.2015

Microbiome

…providing promising targets for potential early detection of CRC…

Furthermore, as we learn more about specific pathways which are associated between the microbiota and CRC, we create more opportunities to uncover potential noninvasive diagnostic targets for this deadly disease.

Mycobiome

As both gut bacteria and fungi independently have the potential to contribute to the inflammatory pathogenesis of CRC, the close interaction between these two entities provides exciting potential to produce unique early diagnostic signatures based on gut bacterial and fungal profiles.

Q

There are many environmental, dietary, and lifestyle factors that influence the microbiome (in both intestine and other tissue), inflammation, immune system, pathogenic mechanisms. The authors should discuss factors other than diet too, eg, smoking, alcohol, obesity, diabetes, bowel habits, etc.

A

Obesity, Smoking and Diabetes

In consideration of lifestyle modifications that also play an adjunctive role in carcinogenesis, studies have now developed further understanding of the role of tobacco smoking in colon cancer development. While studies demonstrate men on average smoke more often than women, the smoking-related colorectal cancer risk is comparable between sexes, with some studies going as far as establishing patterns on higher prevalence on right versus left sided cancers (Gram et al). Current studies have divided pathogenesis into inflammatory responses and modifications in gut microbiome due to smoking. Inflammation caused by smoking has demonstrated changes leading to altered epithelial mucin composition, which in turn disrupts mucus production and enhances the inflammatory response in the gut (Huang et al). Moreover, there is evidence that cigarette smoking leads M2 macrophage polarization and release of IL-1 IL-10, IL-6, IL-8, IL-17, IFN-γ TGF-β1 and TGF-β2, as well as activation of JAK2 and STAT3 (Kim et al; Lee et al; Yuan et al). In regards to microbiome, smoking has demonstrated a negative correlation to the diversity of the gut microbiome. Higher prevalence of Clostridium and Bacteroides in fecal samples of smokers have been identified with lower numbers of Lactococccci, Ruminococcus, and Enterobacteriacae. (Huang et al). Linking together with diet, smoking has demonstrated lower levels of Bifidobacterium, which in turn has reduced the production of protective short-chain fatty acids (Song et al). By this note, studies have also demonstrated smoking cessation to at least partially revive the diversity of the gut microbiome, which may lead to further emphasis on the importance of smoking cessation in colon cancer prevention.

Attention has also been brought to the role of diabetes in colorectal carcinoma development through the lens of inflammation and microbiome modification. Strong links have been established between colorectal cancer and diabetes, with studies also demonstrating worse outcomes when diagnosed (Piper et al). Current studies have established colonic sensory and mechanical changes in diabetes mellitus such as delayed transit time, which alters the gut microbiome composition (Zhao et al). Diabetes mellitus induces histomorphological changes in turn changing gut microbiome function and composition. This affects intestinal immunology, mucosa integrity and smooth muscle function (Zhao et al). Inflammation is a key component of diabetes mellitus with inflammasome regulation of gut microbiota and responses of epithelial cells leading to tissue injury favoring the development of colonic cancer (Zhao et al). More specifically, NFkB has been found to be a key regulator in the crosstalk between diabetes, inflammation and CRC through upregulation of IL-6, IL-1 α and TNF α (Ben-Neriah & Karin et al)). Furthermore, it promotes epithelial to mesenchymal transition through JAK/STAT pathway activation (Ben-Neriah & Karin et al)). Interestingly, TGF-β has also been identified as a key component in the crosstalk between diabetes, inflammation, obesity, and CRC. Consistently elevated glucose levels in type 2 diabetes mellitus leads to upregulation of TGF-β which subsequently leads to M1 macrophage recruitment, B cell apoptosis, reactive oxygen species, and adipocyte hypertrophy (Fischbach et al).  TGF-β also leads to activation of IL-17 producing Th-17 cells which plays an important role in the pathogenesis of colitis (Feagins et al). Given this intricate inflammatory crosstalk, future directions and surveillance has also been considered in combating both colorectal cancer and diabetes together. Studies recently published have considered the role of HMGCoA reductase inhibitors, renin-angiotensin aldosterone system manipulation, vitamin D receptor activators and anti-inflammatory agents in targeting both colorectal cancer and diabetes simultaneously (Piper et al). 

References:

Ben-Neriah, Y., & Karin, M. (2011). Inflammation meets cancer, with NF-κB as the matchmaker. In Nature Immunology (Vol. 12, Issue 8). https://doi.org/10.1038/ni.2060

Feagins, L. A. (2010). Role of transforming growth factor-β in inflammatory bowel disease and colitis-associated colon cancer. In Inflammatory Bowel Diseases (Vol. 16, Issue 11). https://doi.org/10.1002/ibd.21281

Fischbach, S. (2014). The Role of TGF-β Signaling in β-Cell Dysfunction and Type 2 Diabetes: A Review. Journal of Cytology & Histology, 05(06). https://doi.org/10.4172/2157-7099.1000282

Gram, I. T., Park, S. Y., Wilkens, L. R., Haiman, C. A., & le Marchand, L. (2020). Smoking-related risks of colorectal cancer by anatomical subsite and sex. American Journal of Epidemiology, 189(6). https://doi.org/10.1093/aje/kwaa005

Huang, C., & Shi, G. (2019). Smoking and microbiome in oral, airway, gut and some systemic diseases. In Journal of Translational Medicine (Vol. 17, Issue 1). https://doi.org/10.1186/s12967-019-1971-7

Kim, M., Gu, B., Madison, M. C., Song, H. W., Norwood, K., Hill, A. A., Wu, W. J., Corry, D., Kheradmand, F., & Diehl, G. E. (2019). Cigarette smoke induces intestinal inflammation via a Th17 cell-neutrophil axis. Frontiers in Immunology, 10(JAN). https://doi.org/10.3389/fimmu.2019.00075

Lee, G., Jung, K. H., Shin, D., Lee, C., Kim, W., Lee, S., Kim, J., & Bae, H. (2017). Cigarette smoking triggers colitis by ifn-γ+ cd4+ t cells. Frontiers in Immunology, 8. https://doi.org/10.3389/fimmu.2017.01344

Piper, M. S., & Saad, R. J. (2017). Diabetes Mellitus and the Colon. Current Treatment Options in Gastroenterology, 15(4). https://doi.org/10.1007/s11938-017-0151-1

Yuan, F., Fu, X., Shi, H., Chen, G., Dong, P., & Zhang, W. (2014). Induction of murine macrophage m2 polarization by cigarette smoke extract via the JAK2/STAT3 pathway. PLoS ONE, 9(9). https://doi.org/10.1371/journal.pone.0107063

Zhao, M., Liao, D., & Zhao, J. (2017). Diabetes-induced mechanophysiological changes in the small intestine and colon. World Journal of Diabetes, 8(6). https://doi.org/10.4239/wjd.v8.i6.249

Q

There are also influences of germline genetic variations on both immune system and microbiota. Gene-by-environment interactions should be discussed.

A

Given that colorectal cancer risk is governed by both genetic and environmental factors, recent research has focused on delineating these gene-by-environment interactions as potential determinants of CRC development. With respect to gene by environment interactions studied between processed meats and CRC risk, some evidence suggests a modifying effect of the rs4143094 genetic variant on the 10p14/GATA3 gene (Hosoya et al). Although specific mechanisms are not completely understood, given that GATA3 has been shown to be instrumental in T cell development, one potential explanation is that decreased expression of GATA3 and decreased immunogenicity can increase CRC risk in the setting of unopposed inflammation caused by processed meats (Zheng & Blobel et al).

Given that the microbiota plays an important role in the inflammatory pathogenesis of CRC, the genetic-environmental contributions on the gut microbial composition is paramount to our understanding of CRC. In a recently published study, the microbiome of children who were unrelated genetically but living in a similar environment was compared to genetically related children living in different environments (Tavalire et al). Using generalized dissimilarity modeling, it was found that the quantity of a specific microbial taxa was explained by host genetic similarity whereas the species composition was dependent on environmental factors (Tavalire et al). This is especially important in the context of CRC given that certain bacterial species are more likely to be associated with CRC development. For example, Escherichia coli and Bacteroides fragilis have been found in higher quantities in colorectal mucosal tissues from patients with familial adenomatous polyposis that is caused by germline APC mutations (Mima et al).  A genome wide association study for CRC showed that the C2CD2 gene single nucleotide polymorphism was associated with a higher incidence of colorectal adenomas (Chen et al). In a Japanese study, the bacterial family Erysipelotrichaceae, one that is known to produce intestinal inflammation, was found to be associated with the C2CD2 gene single nucleotide polymorphism, suggesting that the C2CD2 gene leads to CRC through the microbiota (Ishida et al). By identifying genetic variants that lead to microbial induced intestinal inflammation and predisposition to CRC formation, we create a unique opportunity for risk stratification and prevention of CRC. In addition, germline genetic variations may play a role in treatment of CRC. In a study which consisted of 361 patients, pathogenic germline variants were found in 15.5% of patients with CRC (Uson et al). 9.4% of patients had findings that would not have been detected by practice guidelines or CRC specific gene testing (Uson et al., 2021). Furthermore, 11% of patients had modifications in their treatment based on pathogenic germline variants (Uson et al., 2021).  

References:

Chen, W., Liu, F., Ling, Z., Tong, X., & Xiang, C. (2012). Human intestinal lumen and mucosa-associated microbiota in patients with colorectal cancer. PLoS ONE, 7(6). https://doi.org/10.1371/journal.pone.0039743

Hosoya, T., Maillard, I., & Engel, J. D. (2010). From the cradle to the grave: Activities of GATA-3 throughout T-cell development and differentiation. Immunological Reviews, 238(1). https://doi.org/10.1111/j.1600-065X.2010.00954.x

Ishida, S., Kato, K., Tanaka, M., Odamaki, T., Kubo, R., Mitsuyama, E., Xiao, J. zhong, Yamaguchi, R., Uematsu, S., Imoto, S., & Miyano, S. (2020). Genome-wide association studies and heritability analysis reveal the involvement of host genetics in the Japanese gut microbiota. Communications Biology, 3(1). https://doi.org/10.1038/s42003-020-01416-z

Mima, K., Kosumi, K., Baba, Y., Hamada, T., Baba, H., & Ogino, S. (2021). The microbiome, genetics, and gastrointestinal neoplasms: the evolving field of molecular pathological epidemiology to analyze the tumor–immune–microbiome interaction. In Human Genetics (Vol. 140, Issue 5, pp. 725–746). Springer Science and Business Media Deutschland GmbH. https://doi.org/10.1007/s00439-020-02235-2

Tavalire, H. F., Christie, D. M., Leve, L. D., Ting, N., Cresko, W. A., & Bohannan, B. J. M. (2021). Shared environment and genetics shape the gut microbiome after infant adoption. MBio, 12(2). https://doi.org/10.1128/mBio.00548-21

Uson, P. L. S., Riegert-Johnson, D., Boardman, L., Kisiel, J., Mountjoy, L., Patel, N., Lizaola-Mayo, B., Borad, M. J., Ahn, D., Sonbol, M. B., Jones, J., Leighton, J. A., Gurudu, S., Singh, H., Klint, M., Kunze, K. L., Golafshar, M. A., Esplin, E. D., Nussbaum, R. L., … Jewel Samadder, N. (2021). Germline Cancer Susceptibility Gene Testing in Unselected Patients With Colorectal Adenocarcinoma: A Multicenter Prospective Study. Clinical Gastroenterology and Hepatology. https://doi.org/10.1016/j.cgh.2021.04.013

Zheng, R., & Blobel, G. A. (2010). Gata transcription factors and cancer. In Genes and Cancer (Vol. 1, Issue 12). https://doi.org/10.1177/1947601911404223

Q

In these lines, research on dietary / lifestyle factors, microbiome, immunity, and personalized molecular biomarkers in tumor is needed for prevention and treatment research. The authors should discuss molecular pathological epidemiology research that can investigate those factors in relation to microbiome, molecular pathologies, immunity, inflammation, and clinical outcomes. Molecular pathological epidemiology research can be a promising direction. Strengths and challenges of molecular pathological epidemiology (in J Pathol 2019, Ann Rev Pathol 2019, Curr Colorectal Cancer Rep 2017, etc.) should be discussed.

A

Another concept which is paramount to our understanding of genetic-environmental implications on the microbiome is molecular pathological epidemiology (MPE). MPE studies are largely based on the principle that human disease is largely heterogeneous due to both intrinsic factors (germline genetic variations, sex) and extrinsic factors (lifestyle, acquired alterations in microbiota)(Hamada et al., 2017). Given this heterogeneity, MPE is able to subclassify patients with a disease into more homogenous groups using molecular pathological markers (Hamada et al., 2017). Using this approach for CRC, we can create more etiologic links between germline genetic variations, and environmental factors with specific subtypes of CRC based on potential immunologic or microbial profiles (Mima et al., 2021). In fact, this has led to recent subfields of MPE including immune-MPE and microbial-MPE (Hamada et al., 2017)

As we have discussed, the immune system plays a considerable role in inflammatory pathogenesis of CRC and is affected by risk factors such as smoking, obesity and diet, and diabetes. Immuno-MPE studies in relation to CRC have demonstrated insight into immunomodulators for CRC immunoprevention (Mima et al., 2021). For example, one study showed that risk reduction of CRC with high FOXP3+ T cell infiltrates was higher when associated with increased intake of marine ω-3 Polyunsaturated fatty acid (ω-3 PUFA), suggesting that marine ω-3 PUFA may create antitumor effects by inhibiting regulatory T cell function (Song et al., 2016). Furthermore, MPE studies have helped link intestinal immunogenicity to the microbiome in the context of CRC. For example, Fusobacterium nucleatum in colorectal tumors was associated with lower levels of tumor infiltrating lymphocytes (TIL) in MSI-high CRC(Hamada et al., 2018). Conversely, it was associated with high level TIL in non MSI-high carcinoma, suggesting that bacteria may play a direct role on tumor-immune microenvironment phenotype(Hamada et al., 2018). In addition, MPE studies have demonstrated that a diet rich in processed and red meat is associated with a higher risk of Fusobacterium nucleatum positive proximal colon cancer, but not with a higher risk of Fusobacterium nucleatum negative proximal cancer(Liu et al., 2018).Meanwhile, a diet rich in whole grains and fiber is associated with a lower risk of Fusobacterium nucleatum associated CRC but not with a lower risk of non-fusobacterium nucleatum associated CRC(Mehta et al., 2017).

Although MPE studies serve as great tool to help with individualized early detection and prognostication of CRC, they still present some challenges. MPE analyses tend to be limited to patients who are able to provide biospecimens which tends to limit study sample size(Hamada et al., 2019). Furthermore, biospecimen collection necessitates a rigorous standardization process of collecting and processing of individual samples(Mima et al., 2021). Since MPE studies often transcend multiple disciplines, multidisciplinary research teams which include fields such as molecular pathology, immunology and microbiology are required(Mima et al., 2021). Finally, given that MPE studies explore heterogeneous diseases, false positive findings may arise given inevitability of multiple hypothesis testing(Hamada et al., 2017). 

In summary, research on genetic-environmental contributions, germline genetic variations, and MPE studies in CRC only reiterates the new, exciting era of medicine which we have entered which focuses on personalized precision medicine. These modalities of studying CRC provide unique opportunities for risk stratification of patients as well as enhanced early detection and prevention of this deadly disease.

References:

Hamada, T., Keum, N. N., Nishihara, R., & Ogino, S. (2017). Molecular pathological epidemiology: new developing frontiers of big data science to study etiologies and pathogenesis. In Journal of Gastroenterology (Vol. 52, Issue 3). https://doi.org/10.1007/s00535-016-1272-3

Hamada, T., Nowak, J. A., Milner, D. A., Song, M., & Ogino, S. (2019). Integration of microbiology, molecular pathology, and epidemiology: a new paradigm to explore the pathogenesis of microbiome-driven neoplasms. In Journal of Pathology (Vol. 247, Issue 5). https://doi.org/10.1002/path.5236

Hamada, T., Zhang, X., Mima, K., Bullman, S., Sukawa, Y., Nowak, J. A., Kosumi, K., Masugi, Y., Twombly, T. S., Cao, Y., Song, M., Liu, L., da Silva, A., Shi, Y., Gu, M., Li, W., Koh, H., Nosho, K., Inamura, K., … Ogino, S. (2018). Fusobacterium nucleatum in colorectal cancer relates to immune response differentially by tumor microsatellite instability status. Cancer Immunology Research, 6(11). https://doi.org/10.1158/2326-6066.CIR-18-0174

Liu, L., Tabung, F. K., Zhang, X., Nowak, J. A., Qian, Z. R., Hamada, T., Nevo, D., Bullman, S., Mima, K., Kosumi, K., da Silva, A., Song, M., Cao, Y., Twombly, T. S., Shi, Y., Liu, H., Gu, M., Koh, H., Li, W., … Giovannucci, E. L. (2018). Diets That Promote Colon Inflammation Associate With Risk of Colorectal Carcinomas That Contain Fusobacterium nucleatum. Clinical Gastroenterology and Hepatology, 16(10). https://doi.org/10.1016/j.cgh.2018.04.030

Mehta, R. S., Nishihara, R., Cao, Y., Song, M., Mima, K., Qian, Z. R., Nowak, J. A., Kosumi, K., Hamada, T., Masugi, Y., Bullman, S., Drew, D. A., Kostic, A. D., Fung, T. T., Garrett, W. S., Huttenhower, C., Wu, K., Meyerhardt, J. A., Zhang, X., … Ogino, S. (2017). Association of dietary patterns with risk of colorectal cancer subtypes classified by Fusobacterium nucleatum in tumor tissue. JAMA Oncology, 3(7). https://doi.org/10.1001/jamaoncol.2016.6374

Mima, K., Kosumi, K., Baba, Y., Hamada, T., Baba, H., & Ogino, S. (2021). The microbiome, genetics, and gastrointestinal neoplasms: the evolving field of molecular pathological epidemiology to analyze the tumor–immune–microbiome interaction. In Human Genetics (Vol. 140, Issue 5, pp. 725–746). Springer Science and Business Media Deutschland GmbH. https://doi.org/10.1007/s00439-020-02235-2

Song, M., Nishihara, R., Cao, Y., Chun, E., Qian, Z. R., Mima, K., Inamura, K., Masugi, Y., Nowak, J. A., Nosho, K., Wu, K., Wang, M., Giovannucci, E., Garrett, W. S., Fuchs, C. S., Ogino, S., & Chan, A. T. (2016). Marineω-3 polyunsaturated fatty acid intake and risk of colorectal cancer characterized by tumor-infiltrating T cells. JAMA Oncology, 2(9). https://doi.org/10.1001/jamaoncol.2016.0605

Round 2

Reviewer 1 Report

The authors have addressed all of my comments.